# Contribution of the Idylla™ System to Improving the Therapeutic Care of Patients with NSCLC through Early Screening of *EGFR* Mutations

Constance Petiteau [1,†], Gwladys Robinet-Zimmermann [2,†], Adèle Riot [3,†], Marine Dorbeau [2], Nicolas Richard [4,5], Cécile Blanc-Fournier [2], Frédéric Bibeau [3], Simon Deshayes [1], Emmanuel Bergot [1,6], Radj Gervais [7] and Guénaëlle Levallet [3,5,6,*]

[1] Department of Pulmonology & Thoracic Oncology, Centre Hospitalier Universitaire de Caen, F-14000 Caen, France; petiteau.co@gmail.com (C.P.); deshayes-si@chu-caen.fr (S.D.); bergot-e@chu-caen.fr (E.B.)

[2] Department of Pathology, Centre François Baclesse, F-14000 Caen, France; gwladys.robinet.zimmermann@chu-rennes.fr (G.R.-Z.); m.dorbeau@baclesse.unicancer.fr (M.D.); c.blanc-fournier@baclesse.unicancer.fr (C.B.-F.)

[3] Department of Pathology, Centre Hospitalier Universitaire de Caen, F-14000 Caen, France; adele.riot@gmail.com (A.R.); fredbibo14@gmail.com (F.B.)

[4] Department of Genetics, Normandy University, UNICAEN, Caen University Hospital, EA 7450 BioTARGen, F-14000 Caen, France; richard-n@chu-caen.fr

[5] Federative Structure of Cyto-Molecular Oncogenetics (SF-MOCAE), Centre Hospitalier Universitaire de Caen, F-14000 Caen, France

[6] Normandy University, UNICAEN, CEA, CNRS, ISTCT Unit, GIP CYCERON, F-14000 Caen, France

[7] Department of Thoracic Oncology, Centre François Baclesse, F-14000 Caen, France; r.gervais@baclesse.unicancer.fr

* Correspondence: guenaelle.levallet@unicaen.fr

† Equal contribution.

**Abstract:** Epidermal growth factor receptor (EGFR) genotyping, a critical examen for the treatment decisions of patients with non-small cell lung cancer (NSCLC), is commonly assayed by next-generation sequencing (NGS), but this global approach takes time. To determine whether rapid *EGFR* genotyping tests by the Idylla™ system guides earlier therapy decisions, *EGFR* mutations were assayed by both the Idylla™ system and NGS in 223 patients with NSCLC in a bicentric prospective study. Idylla™ demonstrated agreement with the NGS method in 187/194 cases (96.4%) and recovered 20 of the 26 (77%) *EGFR* mutations detected using NGS. Regarding the seven missed *EGFR* mutations, five were not detected by the Idylla™ system, one was assayed in a sample with insufficient tumoral cells, and the last was in a sample not validated by the Idylla™ system (a bone metastasis). Idylla™ did not detect any false positives. The average time between *EGFR* genotyping results from Idylla™ and the NGS method was $9.2 \pm 2.2$ working days (wd) ($12.6 \pm 4.0$ calendar days (cd)). Subsequently, based on the Idylla™ method, the timeframe from tumor sampling to the initiation of EGFR-TKI was $7.7 \pm 1.2$ wd ($11.4 \pm 3.1$ cd), while it was $20.4 \pm 6.7$ wd ($27.5 \pm 7.7$ cd) with the NGS method ($p < 0.001$). We thus demonstrated here that the Idylla™ system contributes to improving the therapeutic care of patients with NSCLC by the early screening of *EGFR* mutations.

**Keywords:** non-small cell lung cancer; molecular diagnosis; *EGFR* mutations; NGS; Idylla™

## 1. Introduction

Lung carcinoma remains the most common cause of cancer death worldwide. Nearly 85% of lung cancers are non-small cell lung cancer (NSCLC), and 15% are small cell lung cancer (SCLC). Accounting for approximately 50% of cases of NSCLC, adenocarcinoma (ADC) is the most prevalent histological subtype [1].

All histology combined, the prognosis of patients with NSCLC remains poor, with a five-year survival rate of 15%, NSCLC being most often diagnosed at an advanced (stages III and IV) and metastatic stage [2]. Surgery is therefore insufficient for the therapeutic care of these patients, who can then benefit from chemotherapy based on platinum salts, immunotherapy according to the tumor expression of PDL1 (programmed death-ligand 1), or targeted therapy depending on the molecular abnormalities detected in their tumor. PD-1 blockade alone or with platinum-based chemotherapy is indeed the first-line therapy (depending on the level of PDL1 expression) for non-targetable metastatic NSCLC, while never-smoking patients with NSCLC more often harbor a targetable molecular aberration. The natural history of NSCLC is actually linked to the occurrence of "driver" molecular abnormalities, among which are *KRAS* (24%), *epidermal growth factor receptor* (*EGFR*; 15%), *ERBB2* (<5%), or *BRAF* (<5%) gene mutations; *MET* amplification (<5%); *MET* exon 14 skipping (<5%) and *ALK* (<5%), *ROS1* (<2%), or *RET* (<1%) gene rearrangements [3]. Patients with NSCLC and *EGFR* mutations or *ALK* or *ROS1* gene rearrangement are eligible for first-line targeted therapy with the respective appropriate drug, resulting in greatly improved clinical outcomes [4–6]. The optimal therapy for each patient must be promptly identified to improve patient outcomes in advanced NSCLC. Considering this objective, the mutational profile of NSCLC tumors is essential for developing a more targeted approach in lung cancer treatment. The guidelines recommend searching for *EGFR* (exons 18–21), *ROS1*, *KRAS*, *ALK*, and *PDL1* status in all new diagnosed advanced non-squamous NSCLC and advanced squamous cell carcinomas in never-smokers. The molecular testing turnaround time should not exceed 10 working days according to oncology and pathology societies [7]. Following the ADAURA study, *EGFR* mutation testing is also now recommended in patients with stage IB to IIIA resected NSCLC [8], but with less urgency in terms of the expected timeframe for the result.

Next-generation sequencing (NGS) using dedicated gene panels is the most common approach to detect *EGFR* mutations. This is a global analysis that detects other genomic alterations of interest (*KRAS*, *HER2*, *BRAF*, *MET*, *STK11*, *PIK3CA*, etc.). However, this approach is time consuming. The average expected time of a result of the analysis is ~15 days, when these results are necessary for initiation of treatment of patients. Alternative techniques have been developed, such as the Idylla[TM] *EGFR* mutation assay, i.e., an automated real-time polymerase chain reaction detecting *EGFR* mutations with minimal delays for guiding clinical decisions and starting targeted therapies earlier. This assay detects both *EGFR* mutations (exon 19 deletions, exon 21 (L858R and L861Q), and exon 18 (G719A/C/S) mutations) associated with sensitivity to EGFR-TKIs [9–11] as the main mutations (mainly T790M and insertions in exon 20), predicting the resistance to EGFR-TKIs from the first or second generation but only the response to EGFR-TKIs from the third generation [10,11].

Previous studies have already reported the high concordance (from 94% to 100%) between the Idylla[TM] system and other methods routinely used for *EGFR* mutation detection from formalin-fixed paraffin-embedded (FFPE) tissue of human lung cancer (NGS: [12–22]; pyrosequencing: [23]; Therascreen® *EGFR* RGQ PCR: [24]; droplet digital PCR: [22]; ARMS PCR: [25]; fragment PCR: [26,27]; multiplex PCR: [28]; real-time PCR (cobas® *EGFR* Mutation Test): [29]; Sanger sequencing: [29,30]), usually through a retrospective and monocenter study, although sometimes prospective [13,18,28,29,31] and rarely multicentric [15]. Previous studies have thus evaluated the sensitivity, concordance, and reproducibility between NGS and the alternative technical approaches of *EGFR* genotyping, but whether the early genotyping of *EGFR* can initiate more quickly the treatment of patients with NSCLC is not clearly answered in the literature.

Herein, through a prospective bicentric study, we aimed to assess the benefits of the early diagnosis of *EGFR* mutations in the therapeutic initiation in 225 patients with NSCLC, and thus define this benefit from the patient's point of view by evaluating delays in pathology processing and in initiation of therapy.

## 2. Materials and Methods

### 2.1. Patients from the ID-MUT Study and Paraffin-Embedded Specimens

From January 2019 to August 2020, 223 patients with NSCLC diagnosed by pathologists from Caen University Hospital (CHU; *n* = 79) and the François Baclesse Center (CFB; *n* = 144) were routinely tested to evaluate *EGFR* mutations with both reference methods, i.e., NGS and the Idylla™ system (Table 1).

**Table 1.** Patients' characteristics.

| | Total (*n* = 223) | CHU (*n* = 79) | CFB (*n* = 144) | Patients with *EGFR* Mutation (*n* = 25) |
|---|---|---|---|---|
| **Population** | | | | |
| Male | 126 (56.5%) | 47 (59.5%) | 79 (54.8%) | 8 (32.0%) |
| Female | 97 (43.5%) | 32 (40.5%) | 65 (45.2%) | 17 (68.0%) |
| Age (mean ± SD) years old | 65.4 ± 9.8 | 64.0 ± 9.2 | 66.2 ± 10.1 | 68.5 ± 10.1 |
| (range) years old | [36.5–89.8] | [39.6–83.2] | [36.5–89.8] | [47.3–81.1] |
| **Smoker status [1]** | | | | |
| Never-smokers | 39 (17.5%) | 10 (14.1%) | 29 (20.1%) | 15 (60.0%) |
| Smokers ≤ 10 pack-years | 9 (4.1%) | 2 (2.8%) | 7 (4.9%) | 4 (16.0%) |
| Smokers > 10 pack-years | 167 (74.8%) | 59 (83.1%) | 108 (75.0%) | 6 (24.0%) |
| **Stage** | | | | |
| I (A/B) | 7 (4/3) (3.1%) | 4 (2/2) (5%) | 3 (2/1) (20.1%) | 1 (1/0) (4%) |
| II (A/B) | 6 (2/4) (2.7%) | 3 (1/2) (3.8%) | 3 (1/2) (20.1%) | 1 (1/0) (4%) |
| III (A/B/C) | 31 (18/10/3) (13.9%) | 8 (4/4/0) (10%) | 23 (14/6/3) (15.9%) | 2 (2/0/0) (8%) |
| IV (A/B) | 164 (47/117) (73.5%) | 52 (10/42) (65.8%) | 112 (37/75) (77.8%) | 21 (7/14) (84%) |
| **Histology according to the WHO 2015 4th edition** | | | | |
| Adenocarcinoma (ADC) | 164 (73.6%) | 57 (72.1%) | 107 (74.3%) | 22 (88.0%) |
| ○ Mucinous subtype | ○ 1 | ○ 1 | ○ 0 | ○ 0 |
| ○ Papillary subtype | ○ 1 | ○ 1 | ○ 0 | ○ 0 |
| ○ Enteric subtype | ○ 3 | ○ 3 | ○ 0 | ○ 0 |
| NSCLC [2] NOS [3] | 30 (13.4%) | 12 (15.2%) | 18 (12.5%) | 0 |
| NSCLC in favor of an ADC | 21 (9.4%) | 10 (12.6%) | 11 (7.5%) | 3 (12.0%) |
| Squamous cell carcinoma | 6 (2.7%) | 0 | 6 (4.1%) | 0 |
| Carcinoid tumor | 1 (0.4%) | 0 | 1 (0.7%) | 0 |
| Small cell carcinoma and compound ADC | 1 (0.4%) | 0 | 1 (0.7%) | 0 |
| **Nature of tumoral sample** | | | | |
| Fibroendoscopy biopsy | 87 (39.0%) | 42 (53.2%) | 45 (31.2%) | 9 (36.0%) |
| Fine needle aspiration | 57 (25.6%) | 30 (38.0%) | 27 (18.8%) | 6 (24.0%) |
| Biopsy by scanner | 42 (18.8%) | 0 | 42 (29.2%) | 6 (24.0%) |
| Ultrasound biopsy | 30 (13.5%) | 2 (2.5%) | 28 (19.4%) | 3 (12.0%) |
| Surgical biopsy | 7 (3.1%) | 5 (6.3%) | 2 (1.4%) | 1 (4.0%) |
| **Localization** | | | | |
| Lung | 112 (50.2%) | 47 (59.5%) | 65 (45.2%) | 12 (48.0%) |
| Lymphadenopathy | 61 (27.3%) | 27 (34.2%) | 34 (23.7%) | 6 (24.0%) |
| Bone metastasis | 20 (8.9%) | 1 (1.3%) | 19 (13.0%) | 5 (20.0%) |
| Pleural metastasis | 5 (2.3%) | 1 (1.3%) | 4 (2.8%) | 0 |
| Metastasis in other locations | 21 (9.4%) | 0 | 21 (14.6%) | 1 (4.0%) |
| Brain | 2 (0.9%) | 2 (2.5%) | 0 | 0 |
| Pleural fluid | 2 (0.9%) | 1 (1.3%) | 1 (0.7%) | 1 (4.0%) |

[1] Information missing for 8 patients; [2] NSCLC, non-small lung cancer; [3] NOS, not otherwise specified.

All tumor samples were fixed in formalin for 6 to 48 h: (i) "biopsy" type samples (including bone samples) were addressed to the pathology department from the Caen University Hospital or from the CFB in formalin buffered at 4%, (ii) the "fine needle aspiration" type samples were addressed to the pathology departments in a fixing liquid (CytoRich^TM, BD, Le Pont de Claix, France); the cells were pelleted by centrifugation then incubated in 4% buffered formalin. Following the fixation step, the bone samples undergwent an additional decalcification step by being incubated in a buffered EDTA solution (Osteosoft^TM, Merck, Germany). After fixation (and decalcification for bone samples), the tumor samples were embedded with paraffin. Sections of paraffin-embedded specimens for *EGFR* mutation testing by NGS method were performed before those for *EGFR* mutation testing by the Idylla^TM system. At the end of the two series of sections, a slide for hematoxyllin, eosin, saffron (HES) staining and morphological verification of the residual tumor material was systematically carried out.

The clinical data were recovered from electronic medical records, including the date of (1) tumor sampling, (2) multidisciplinary consultation meetings, (3) diagnosis announcement, and (4) initiation of anti-EGFR therapy.

Specific informed consent was obtained for the biological study (ID-MUT). The additional *EGFR* analysis by Idylla^TM was approved by the appropriate ethics committee (CPP Ref DC 2008-574 Nord-Ouest III, France and Local Health Research Ethics Committee (CLERS ref ID#1595), France). In accordance with the French law n° 2018-493 of 20 June 2018 relating to the protection of personal data, a formally complete declaration file was sent to the National Commission for Data Protection (declaration number: 2204611 v 0).

### 2.2. EGFR Mutation Assay by Next-Generation Sequencing Panel CLv3 (Colon and Lung Cancer Panel v3)

*EGFR* mutation testing by NGS method was centralized and carried out once a week in the Department of Genetics from the CHU de Caen. Tumor genomic DNA was extracted from three sections of 10 μm of the paraffin-embedded specimens using an RSC FFPE Plus DNA Kit (Promega^TM, Charbonnières-les-Bains, France) on a Maxwell RSC 48 automated system (Promega^TM) according to the manufacturer's recommendation twice a week. The DNA concentration was measured with a NanoDrop 2000 spectrophotometer (Thermo Fisher Scientific, Waltham, Massachusetts, USA).

NGS was performed using S5 Prime (Thermo Fisher Scientific). The average depth was $>500\times$; on target $>90\%$. Bioinformatic analyses (alignment, call of variants, and annotations) was run on LifeTechnologies: Torrent suite 5.10, Variant caller 5.10, Ion reporter 5.10—Nextgene (Softgenetics, State College, Pennsylvania, USA) 2.4.1.2. The copy number variant (CNV) analysis was expressed as the ratio of mean depths by amplicons $\pm 2$ standard deviations. The detection limit was set to 3% for punctual mutations and 5% for insertions/deletions for a minimum depth of $100\times$ per amplicon. Variations of the sequences recognized as non-pathogenic (classes 1 and 2) were not mentioned. The allelic frequency of variants (VAF) of an alteration was evaluated, including panel CLv3 sequence exons 18–21 of the *EGFR* gene.

### 2.3. EGFR Mutation Assay by the Idylla^TM System

*EGFR* mutation testing by the Idylla^TM system was centralized and carried out every working day in the Department of Pathology from the CHU de Caen from unextracted paraffin-embedded specimens according to an adaptation of the manufacturer's recommendation. Briefly, three sections of 20 μm thickness from the paraffin-embedded specimens were loaded into the Idylla^TM *EGFR* Mutation cartridge for the following fully automated test, previously described by others [32]. The tumor sample had to contain at least 10% of tumor cells; a macrodissection was carried out to enrich the sample with tumor cells when necessary. The *EGFR* mutations detected in the Idylla^TM *EGFR* Mutation cartridge are listed in Supplementary Table S1.

PCR curves were visualized through the web-based interphase Idylla^TM explore to evaluate the quantification cycle (CQ) from the mutation signal if present, the CQ for the internal control (total *EGFR*) signal, the difference between the two CQs (ΔCQ), the sigmoid aspect of the amplification curve of the mutation when found, and the maximal fluorescence, similarly to Momeni-Boroujeni et al. [32]. The total *EGFR* CQ for all samples ranged from 16 to 26.

### 2.4. Statistical Analysis

Sensitivity and specificity were the proportion of concordant results against the sum of concordant and discordant results (true positives/(true positives + false negatives) and true negatives/(true negatives + false positives), as detailed by [24]. The male/female distribution, smoking status, and age of patients between NSCLC patients with or without *EGFR* mutation were evaluated using a chi-square test and a non-parametric test for unpaired data from Mann–Whitney. The influence of the used method (Idylla^TM or NGS) on the time required for delivering *EGFR* genotyping results was evaluated by a two-way (techniques and time) analysis of variance (ANOVA), followed by a post-hoc Bonferroni test (GraphPad Prism version 8.0.0 Software, San Diego, California, USA).

Statistical differences of the timeframe according to the method used for *EGFR* genotyping (Idylla^TM or NGS) were determined using a *t*-test (GraphPad Prism version 8.0.0 Software, San Diego, California, USA). Statistical significance was set at $p \le 0.05$.

### 3. Results

#### 3.1. Patients' Characteristics

In total, 223 patients (126 men and 97 women; medium age at diagnosis: 65.4 ± 9.8 years old) newly diagnosed with NSCLC, mostly adenocarcinomas or probable adenocarcinomas (83.1%), were enrolled in the ID-MUT study from January 2019 to August 2020 (Table 1, Figure 1).

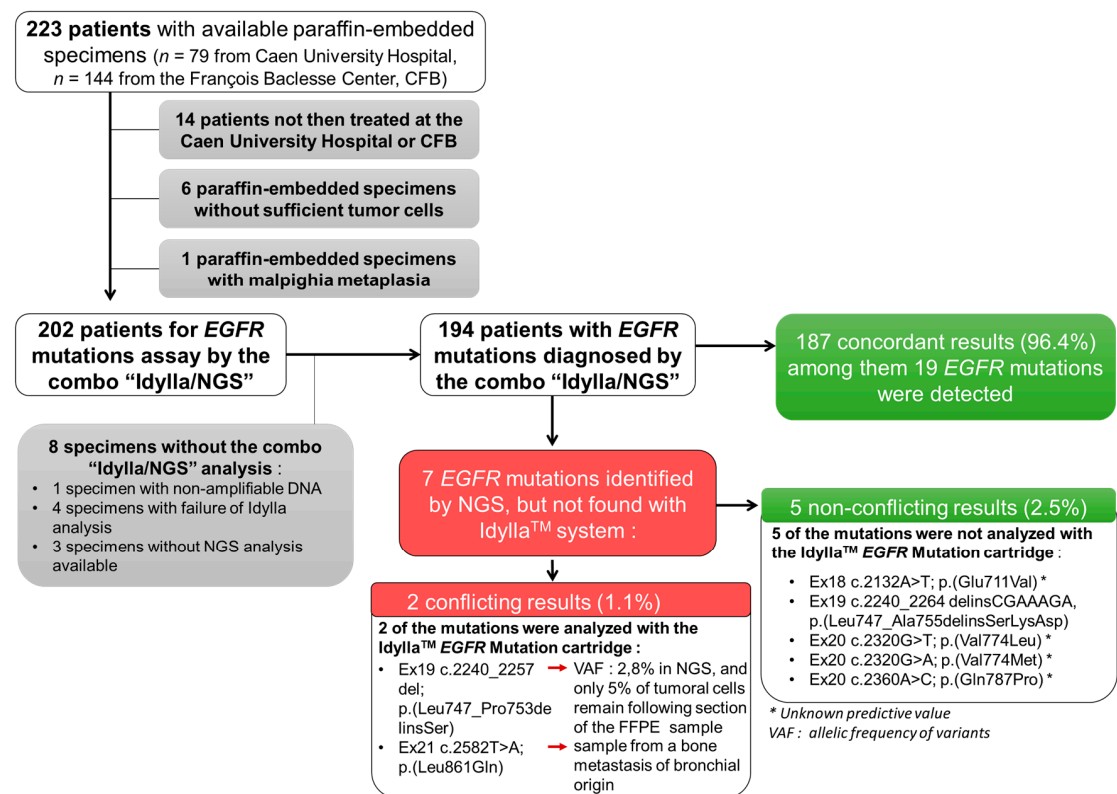

**Figure 1.** Flow chart of the ID-MUT study and concordance between the Idylla^TM system and NGS genotyping for *EGFR*.

We excluded from this study: (i) patients not then treated at the CHU or CFB (*n* = 14), (ii) samples with insufficient material (tumor sample of less than 50 mm$^2$ and/or without sufficient tumor cells (less than 10% of tumoral cells); *n* = 6), (iii) a malpighia metaplasia sample (*n* = 1), and (iv) samples without results for both the Idylla$^{TM}$ and NGS tests (*n* = 8) (Figure 1).

The final analysis was thus performed on 194 NSCLC samples (Figure 1). Of the 194 NSCLC samples tested, 149 (76.8%) were histological FFPE specimens and 45 (23.2%) were cytology specimens.

### 3.2. Concordance of EGFR Genotyping between the Idylla$^{TM}$ and NGS Methods

Of the 194 NSCLC samples tested, 25 (12.8%) were positive for *EGFR* mutations (including a sample with a double mutation (EGFR_ex20 c.2360A > C; p. (Gln787Pro) (Q787P)/EGFR_ex20 c.2369C > T; p. (Thr790Met) (T790M)) and 169 negatives (87.2%) using NGS, the benchmark analysis (Figure 1). Among these 25 patients were a majority of women (*n* = 17 (68.0%); *p* < 0.01) and a majority of non-smokers (*n* = 15, (60%); *p* value < 0.001). With Idylla$^{TM}$, 19 (9.7%) and 175 (90.3%) cases were positive and negative, respectively, for *EGFR* mutations (Figure 1). Thus, Idylla$^{TM}$ demonstrated agreement with the NGS method in 187/194 cases (96.4%) and recovered 20 of the 26 (74.1%) *EGFR* mutations detected using NGS: 11 deletions in exon 19, seven L858R mutations, one T790M mutation, and one insertion in exon 20 from *EGFR*. In addition, Idylla$^{TM}$ did not detect any false positives.

However, seven *EGFR* mutations were detected by NGS but not by Idylla$^{TM}$; five of these missed mutations were not assayed by the Idylla$^{TM}$ system and were therefore not true mismatches between Idylla$^{TM}$ and NGS. The two other missed mutations were mutations evaluated by Idylla$^{TM}$. However, one of them was missed probably because, following the scraping of the FFPE block for Idylla$^{TM}$ to analyze, only 5% of the tumor cells remained in the sample (the sensitivity threshold of the Idylla$^{TM}$ technique is 10% of tumor cells), and the deletion of the exon 19 of the EGFR reported by NGS had a low allelic frequency of 2.80%. The second missed mutation was from a bone metastasis sample of an undifferentiated carcinoma presumed to be of pulmonary origin, because it was TTF1-positive. It should be noted that the Idylla$^{TM}$ system has not been certified for such type of sample.

### 3.3. Consideration of the EGFR Genotyping by the Idylla$^{TM}$ Method in the Treatment Decision

Among the 194 patients with *EGFR* mutations diagnosed by the combo "Idylla$^{TM}$/NGS", 158 patients received first-line systemic treatment. For these 158 patients, *EGFR* genotyping results by Idylla® were all reported before those by NGS. For the majority of these patients, i.e., 118/158 of them (75%), the multidisciplinary consultation meeting leading to the therapeutic decision took place with the knowledge of *EGFR* genotyping by the Idylla$^{TM}$ method and without the knowledge of the result of the analysis by NGS. For 23/158 patients, *EGFR* genotyping results were known (10/23 *EGFR* genotyping results were based only on the Idylla$^{TM}$ method, 13/23 *EGFR* genotyping results were known from both NGS and Idylla$^{TM}$ methods). Finally, 17/158 patients (11%) were discussed in the multidisciplinary consultation meeting before the results of the mutation status of *EGFR*.

### 3.4. Turnaround Time (TAT)

To appreciate the TAT of *EGFR* genotyping from the tumor sample to the initiation of EGFR-TKI treatment according to the NGS or Idylla$^{TM}$ method, we then measured the timeframe between (1) tumor sampling and *EGFR* genotyping request, (2) *EGFR* genotyping request and result, (3) tumoral sampling and *EGFR* genotyping result, (4) results from both techniques, and (5) tumoral sampling and initiation of treatment (Figure 2).

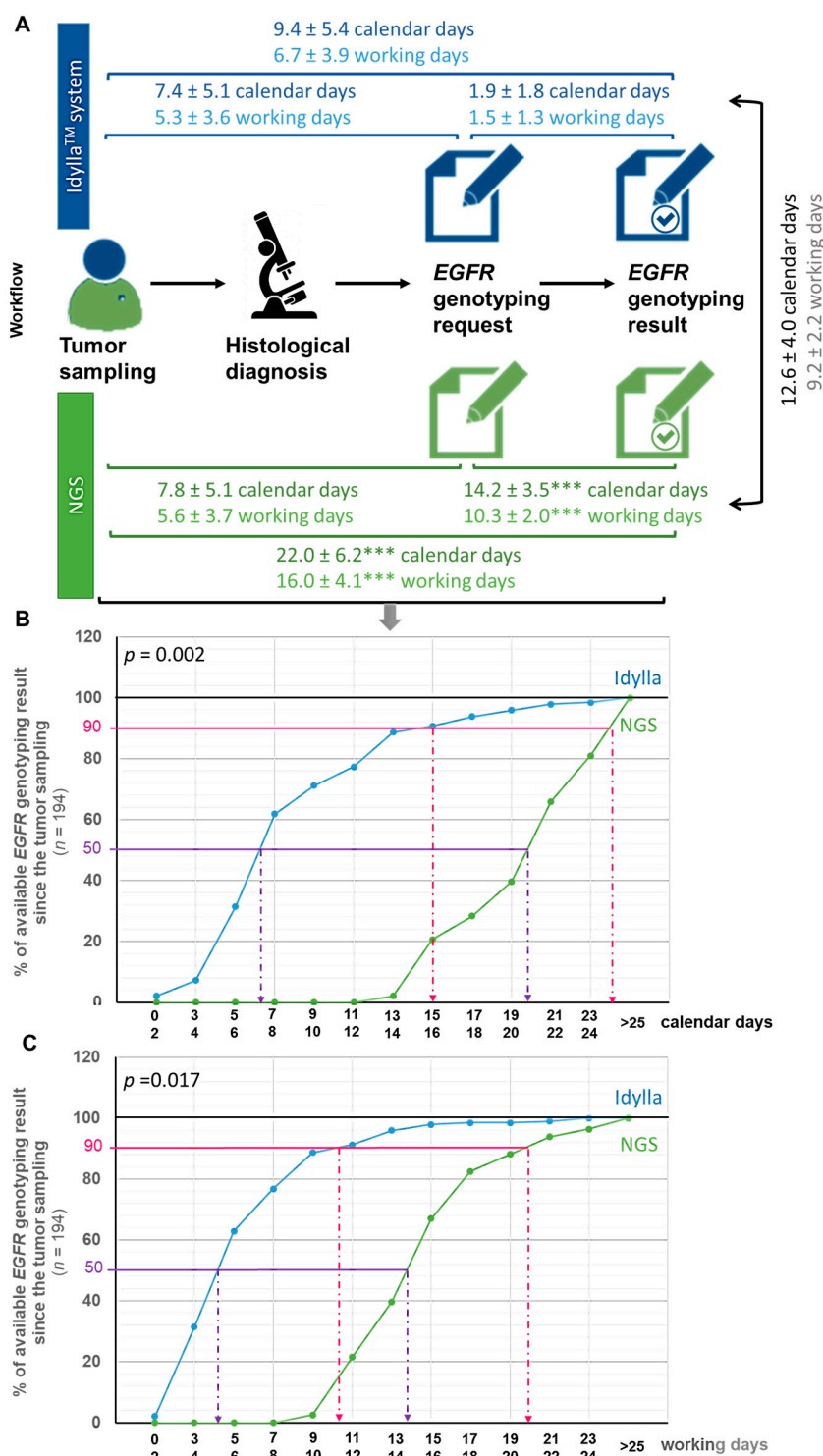

**Figure 2.** Turnaround time from the tumor sample to the initiation of treatment according to the NGS or Idylla$^{TM}$ method. (**A**) Workflow illustrating delays between each step toward the communication of *EGFR* genotyping result in calendar or working days (mean ± SD). Delays according to the Idylla$^{TM}$ system or NGS assay was tested using a *t*-test (\*\*\* $p < 0.001$ when compared in the same timeframe to the Idylla$^{TM}$ method). (**B**,**C**) Monitoring the availability of *EGFR* genotyping results by the Idylla$^{TM}$ or NGS technique from tumor sampling in calendar days (**B**) or working days (**C**). The influence of the "technique" variable (Idylla$^{TM}$ or NGS) on the time required to deliver *EGFR* genotyping result was evaluated by a two-way (techniques and time) analysis of variance (ANOVA), followed by a post-hoc Bonferroni test (GraphPad Prism version 8.0.0 Software, San Diego, CA, USA).

We expressed these time periods in both working (wd) and calendar (cd) days, calendar days being more representative for patients.

The TAT from tumor sampling to *EGFR* genotyping request was comparable whether genotyping was ordered from Idylla$^{TM}$ or performed using the NGS method, with a timeframe of 5.3 ± 3.6 wd (7.4 ± 5.1 cd) and 5.6 ± 3.7 wd (7.8 ± 5.1 cd), respectively (Figure 2A). By contrast, the TAT from the *EGFR* genotyping request to the *EGFR* genotyping results was almost six times faster with Idylla$^{TM}$ than with the NGS method, with a timeframe of 1.5 ± 1.3 wd (1.9 ± 1.8 cd) and 10.3 ± 2.0 wd (14.2 ± 3.5 cd), respectively ($p < 0.001$) (Figure 2A). The average time between *EGFR* genotyping results by Idylla$^{TM}$ and the NGS method was thus 9.2 ± 2.2 wd (12.6 ± 4.0 cd) (Figure 2A). All these TAT (timeframe between (1) tumor sampling and EGFR genotyping prescriptionrequest, (2) EGFR genotyping prescription request and result, (3) tumoral sampling and EGFR geno-typing result, (4) results from both techniques, and (5) tumoral sampling and initiation of treatment) were comparable between the CHU and the CFB ($p > 0.05$).

Figure 2B,C illustrates the availability of *EGFR* genotyping since the tumor sampling in calendar days (Figure 2B) and working days (Figure 2C) from the point of view of patients (calendar days) and practitioners (working days), respectively. As shown, for half of the patients from the ID-MUT study, the *EGFR* genotyping result was determined to be 6–7 cd (4–5 wd) after the tumor sampling with the Idylla$^{TM}$ method, against 19–20 cd (13–14 wd) for the NGS method. Similarly, for 90% of the patients from the ID-MUT study, the *EGFR* genotyping result was determined to be 15–16 cd (10–11 wd) after tumor sampling with the Idylla$^{TM}$ method, against 23–24 cd (19–20 wd) for the NGS method. The influence of the "technique" used (Idylla$^{TM}$ or NGS) on the time required to deliver *EGFR* genotyping results was evaluated by a two-way (technique and time) analysis of variance (ANOVA), followed by a post-hoc Bonferroni test, which confirmed statistically the time-saving benefit of the Idylla$^{TM}$ technique whether evaluated in cd ($p = 0.002$) or wd ($p = 0.0017$).

During the first part of the ID-MUT study (the first nine months), in one of the centers (Caen University Hospital), the *EGFR* genotyping results by both the Idylla$^{TM}$ and NGS methods were expected to initiate EGFR-TKI treatment, the time for the confirmation of *EGFR* genotyping concordance between the two methods in our hand. During the second part of this study, EGFR-TKI treatment was initiated on *EGFR* genotyping by Idylla$^{TM}$ in both Caen University Hospital and the François Baclesse Center. For this reason, among the 22 NSCLC patients from the ID-MUT study with *EGFR* mutations treated by EGFR-TKI, 12 NSCLC patients harboring *EGFR* mutations received treatment following *EGFR* genotyping by the NGS method, while the other 10 patients with NSCLC and *EGFR* mutations received EGFR-TKI treatment according *EGFR* genotyping by the Idylla$^{TM}$ system (i.e., before the *EGFR* genotyping result by NGS was known), allowing the evaluation of the contribution of the Idylla$^{TM}$ system to improving the therapeutic care of patients with NSCLC by early screening of *EGFR* mutations (Table 2). Table 2 details the timeframes for the available *EGFR* genotyping according to the Idylla$^{TM}$ or NGS method for each of the 22 patients, as well as the time required for the initiation of EGFR-TKI treatment since the tumor sample was processed.

The time to initiation of EGFR-TKI was defined as the time between the interventional procedures leading to the histological confirmation until initiation of TKIs for patients harboring *EGFR* mutations. The TAT from tumor sampling to initiation of EGFR-TKI was 7.7 ± 1.2 wd (11.4 ± 3.1 cd) when the decision was based on the Idylla$^{TM}$ method, while it was 20.3 ± 6.7 wd (27.2 ± 8.3 cd) when the decision was based on the NGS method, i.e., reduced by more than two-fold with the Idylla$^{TM}$ system ($p < 0.001$).

**Table 2.** Turnaround time of initiation of EGFR-TKI in patients with NSCLC harboring *EGFR* mutations according to *EGFR* genotyping by the Idylla™ system or the NGS method.

| Patient | Gender | Smoker Status Histology p. Stage | *EGFR* Mutation | *EGFR* Genotyping by Idylla™ * | *EGFR* Genotyping by NGS * | EGFR-TKI Initiation * |
|---|---|---|---|---|---|---|
| | | | | **Time for:** | | |
| 18 | F | Never-smoker NSCLC in favor of an ADC, IVB | ex19 c.2240_2257del; p (Leu747_Pro753delinsSer) | 5 cd | 15 cd | 12 cd on Idylla™ |
| 32 | F | Never-smoker ADC, IVB | ex21 c.2573T > G p. (Leu858Arg) | 11 cd | 20 cd | 18 cd on Idylla™ |
| 40 | F | ≤10 pack-years ADC, IVB | ex19 c.2235_2249del; p. (Glu746_Ala750del) | 5 cd | 15 cd | 8 cd on Idylla™ |
| 81 | F | Never-smoker ADC, IVB | ex21 c.2573T > G; p. (Leu858Arg) | 9 cd | 23 cd | 9 cd on Idylla™ |
| 96 | F | Never-smoker ADC, IVB | ex19 c2239_2248delinsC; p. (Leu747_Ala750delinsPro) | 7 cd | 17 cd | 14 cd on Idylla™ |
| 171 | F | >10 pack-years ADC, IVA | ex19 c2240_2254del; p. (Leu747_Thr751del) | 6 cd | 21 cd | 12 cd on Idylla™ |
| 183 | F | Never-smoker ADC, IIIA | ex19 c2235_2249del; p. (Glu746_Ala750del) | 7 cd | 23 cd | 10 cd on Idylla™ |
| 187 | F | ≤10 Pack-years ADC, IVB | ex21 c.2573T > G; p. (Leu858Arg) | 13 cd | 21 cd | 9 cd on Idylla™ |
| 193 | F | Never-smoker ADC, IVA | ex21 c.2573_2579delinsGGGCCAT; p. (Leu858_Lys860delinsArgAlaIle) | 5 cd | 13 cd | 11 cd on Idylla™(1) |
| | | | **Mean in EGFR-TKI Initiation on Idylla™ ± SD: 11.4 ± 3.1 cd (7.7 ± 1.2 wd)** | | | |
| 16 | F | Never-smoker ADC, IVB | ex19 c.2235_2249del; p. (Glu746_Ala750del) | 12 cd | 23 cd | 28 cd on NGS |
| 20 | F | Never-smoker ADC, IVB | ex21 c.2573T > G; p. (Leu858Arg) | 16 cd | 23 cd | 45 cd on NGS |
| 62 | M | Never-smoker ADC, IVB | ex19 c2240_2254del p. (Leu747_Thr751del) | 13 cd | 26 cd | 31 cd on NGS |
| 84 | M | >10 pack-years ADC, IVB | ex19 c2240_2254del; p. (Leu747_Thr751del), eX20 c.2305G > A; p (Val769MET) | 3 cd | 16 cd | 25 cd on NGS |
| 85 | M | Never-smoker ADC, IVA | ex19 c2236_2250del; p. (Glu746_Ala750del) | 7 cd | 19 cd | 25 cd on NGS |
| 97 | F | Never-smoker ADC, IIIA | ex19 c.2235_2249del; p. (Glu746_Ala750del) | 7 cd | 16 cd | 20 cd on NGS |
| 105 | F | Never-smoker ADC, IVB | ex21 c.2582T > A p. (Leu861Gln) | 12 cd | 22 cd | 28 cd on NGS |
| 110 | F | Never-smoker ADC, IVA | ex18 c.2132A > T; p. (Glu711Val) | 5 cd | 15 cd | 15 cd on NGS |
| 134 | M | ≤10 pack-years ADC, IVB | ex20 c.2319_2320insTAC; p. (His773_Val774insTyr) | 8 cd | 22 cd | 29 cd on NGS |
| 162 | F | ≤10 pack-years NSCLC in favor of an ADC, IVB | ex 19 c.2240_2257del; p. (Leu747_Pro753delinsSer) | 7 cd | 16 cd | 20 cd on NGS |
| 194 | M | >10 pack-years ADC, IVB | ex20 c.2320G > A; p. (Val774Met) | 4 cd | 11 cd | 14 cd on NGS |
| 211 | F | Never-smoker ADC, IIA | ex21 c.2573T > G; p. (Leu858Arg) | 13 cd | 16 cd | 27 cd on NGS |
| | | | **Mean in EGFR-TKI Initiation on NGS ± SD: 27.2 ± 8.3 cd (20.3 ± 6.7 wd)** | | | |

* In calendar days (cd) or working day (wd) from tumor sampling. (1) detected, although not on mutations detected with Idylla™ EGFR Mutation cartridge, by Idylla™ method as ex21 c.2573T > G; p. (Leu858Arg) mutation.

## 4. Discussion

In this study, we confirmed the good sensitivity and specificity of the rapid detection of *EGFR* mutations using the Idylla™ system and mainly reported that *EGFR* mutation detection with this assay is associated with a significantly reduced turnaround time compared

to the use of NGS testing. In turn, patients were observed to begin systemic EGFR-TKIs therapy an average of two weeks earlier than if waiting for the NGS result.

The good sensitivity and specificity of the rapid detection of *EGFR* mutations using the Idylla^TM system were previously reported in other publications on lung cancer, as well as on other cancers such as melanoma and colorectal cancer [17]. Similarly, as detailed in the introduction, the concordance of *EGFR* genotyping has already been reported between the Idylla^TM system and NGS [12–22] or other techniques [23–30]. In our center, we chose to introduce the Idylla^TM system rather than another rapid assay, because our main objective was to reduce, at maximum, the result of the *EGFR* genotyping for clinicians and to reduce the need of DNA extraction being performed daily, not allowing this objective to be achieved. In fact, the Idylla^TM system is one of the rare solutions avoiding DNA extraction and allowing reliable *EGFR* genotyping directly from FFPE sample slides [20,25,33–35] in all patients with advanced NSCLC, mainly adenocarcinomas and squamous cell carcinomas in never-smokers, as in patients with stage IB to IIIA resected *EGFR* mutation-positive NSCLC [8]. However, the Idylla^TM system has several limitations. First, biological materials obtained from biopsies are often limited and can be an issue when multiple analyses are required. Indeed, the Idylla^TM assay requires additional sections of FFPE sample and therefore risks exhausting the tumor sample, especially because, contrary to what is recommended by Biocartis, which markets the Idylla^TM *EGFR* cartridge, we did not perform *EGFR* genotyping on a single section of 5 μm but on three sections of 20 μm thickness from the paraffin-embedded specimens, because, in our hands, during preliminary tests, we observed that, by following the recommendations of Biocartis, there was a risk of missing out on an *EGFR* mutation with low allelic frequency. This did not put the performance of the tests at risk because no analysis failure due to a saturation of a cartridge was reported. As we have an excellent agreement (96.4%) of results between the methods by the Idylla^TM system and NGS, we concluded that our procedure for *EGFR* mutation testing by the Idylla^TM system allowed us not to miss a mutation due to lack of sensitivity for the Idylla^TM method. The off-label use of CE-IVD methods was thoroughly validated before being used in routine testing during a retrospective study not reported here. The risk to exhaust the tumor sample could, however, be lifted by the reuse of H&E, immunohistochemistry (IHC), and fluorescence in situ hybridization (FISH) diagnostic slides [36], by the use of plasma from patients with lung cancer [37,38] or of the DNA extracted from FFPE sections for NGS analysis [14,19,26,28,39]. Indeed, using the same DNA for Idylla^TM and NGS assays could discard divergent results linked to tumor heterogeneity and to the fact that the two analyses are not carried out on strictly the same part of the tumor sample. Moreover, only a little DNA is needed; Bocciarelli et al. reported that >25 ng of DNA and >10% of tumor cells are sufficient to detect *EGFR* mutations with the Idylla^TM method [19]. DNA use seems a good alternative, however, and this is also the second limitation of the Idylla^TM system—Idylla^TM *EGFR* cartridges are not certified for samples other than primary tumor biopsies included in FFPE. This lack of certification is regrettable, but like others [13,27,30,40], we also analyzed samples of other kinds (fine needle aspiration and metastasis of bronchial origin) on Idylla^TM *EGFR* cartridges and, except for one of them, the results of *EGFR* genotyping were consistent with the analysis by NGS. As long as the result of the *EGFR* genotyping is confirmed in a second step by another technique, it seems to us that this second limitation can therefore be avoided.

Studies comparing the Idylla^TM system and NGS performance [12–22] reported a concordance between the two techniques ranging from 94% to 100%, which is consistent with the 96.7% of concordant results between the Idylla^TM system and NGS that we evaluated here, especially because the seven apparent discordant results that we reported were not trues discordant: Five missed mutations were not detected by the Idylla^TM system, one missed mutation was assayed in a sample with insufficient tumoral cells, and the last missed mutation was sought in a sample not validated on the Idylla^TM system (a bone metastasis).

That some mutations detected in the NGS panel were absent in the Idylla[TM] test panel is unfortunate, but most of the time there are few consequences for the patient considering that approved therapies are missing for most of those missed mutations (insertion in exon 20 of the *EGFR* gene), and the others were finally found with the complementary analysis of the sample by NGS, such as the C797S mutation, a second acquired resistance mutation arising in tumors that have progressed after treatment for T790M+ disease and not detected in the Idylla[TM] system [41]. For us, the Idylla[TM] system or another rapid system of *EGFR* mutation detection is essential for reducing the timeframe of *EGFR* genotyping and initiating therapy in patients with lung cancer. However, it cannot be the only analysis realized to process this genotyping because of the risk of missing some *EGFR* mutations, either in samples with few tumor cells or rare *EGFR* mutations not detected by rapid genotyping, but for which we will soon know whether they do or do not predict the response to EGFR-TKIs, and because NGS allows the analysis of a large panel of genes whose alterations (mutations and copy gain) can also guide the treatment decisions of patients with lung cancer. Considering the simultaneous evaluation of numerous genomic alterations across several genes with NGS, and even if the system is presented as being available in any laboratory, because it does not require a molecular biologist, we believe that the links between these laboratories and platforms equipped with NGS technology must be preserved. NGS panels remain essential in molecular sub-type diagnosis of lung cancer and cannot be replaced due to the rapid emergence of new targeted therapies for different genomic alterations. Therefore, it can screen mutations that allow some patients to be included into clinical trials. Molecular testing is also essential in the treatment strategy because studies have demonstrated that immunotherapy before targeted therapies increases the occurrence of serious side effects [42–44]. While comprehensive molecular screening is essential in academic centers with access to clinical trials, it is questionable in smaller centers who do not have access to NGS assays. In those centers, Idylla[TM] assays can be part of the solution to improving the time to initiate therapies.

Molecular testing requires a good-quality sample, enough tumor cells, and even multiple interventional procedures to be conclusive, which would lengthen delays. In 2016, a survey from the French National Cancer Institute showed that the median turnaround time (TAT) from test prescription to reception of results by the clinician for *EGFR* molecular test was 18 days [7]. A limitation with the NGS technique is that the TAT is usually longer than the TAT associated with a specific assay. However, a delayed turnaround time for biomarker reports can lead to delays in treatment initiation, decreased efficacy of treatment, and inappropriate treatment decisions. Indeed, *EGFR* mutations are an oncogenic driver occurring especially in patients with pulmonary adenocarcinoma and never-smokers. For patients with metastatic NSCLC harboring *EGFR* mutations, the development of EGFR tyrosine kinase inhibitors (TKIs) is an important improvement in therapeutic care, as shown by the increase in the progression-free survival (PFS) and limitation of toxicities using EGFR-TKIs for patients with *EGFR*-mutated NSCLC compared to chemotherapy [5], especially with the use of osimertinib versus first-line TKIs (gefitinib and erlotinib) [4,6]. However, a long delay to initiate EGFR-TKIs can result in rapid disease progression and deterioration in performance status associated with a worse prognosis [45]. Thus, it is essential to accelerate the availability of molecular sub-type results. Because pathology processing is reduced with rapid techniques, we were able obtain a result for *EGFR* genotyping 12.5 calendar days earlier with Idylla[TM] compared to NGS assays. As a consequence, the TAT from tumor sampling to initiation of EGFR-TKIs was reduced by two weeks when the decision was based on the Idylla[TM] method compared to when the decision was based on the NGS method. Besides faster delivery of the appropriate treatment for patients with NSCLC, therefore increasing their chance of survival, we can also assume that by improving the deadlines, we can improve patients' adherence to participating in clinical trials.

## 5. Conclusions

This study demonstrated, for the first time and to the best of our knowledge, the benefit for the patient of the introduction into routine practice of the rapid *EGFR* genotyping test, in addition to NGS in the initiation of its therapeutic care. *EGFR* mutation assays by the Idylla^TM system, in addition to NGS testing, increase the costs of patient care but improve it through the timely completion of biomarker results and the facilitation of appropriate treatment decisions [46].

**Supplementary Materials:** The following are available online at https://www.mdpi.com/article/10.3390/curroncol28060376/s1, Table S1: Mutations detected with the Idylla^TM *EGFR* Mutation cartridge.

**Author Contributions:** Conceptualization, F.B., E.B., R.G. and G.L.; methodology, C.P., A.R. and G.L.; software, C.P., A.R. and G.L.; validation, G.L.; formal analysis, C.P., A.R. and G.L.; investigation, C.P., G.R.-Z. and A.R.; resources, C.P., G.R.-Z., M.D., C.B.-F., S.D. and A.R.; data curation, C.P., G.R.-Z., A.R., M.D., C.B.-F., S.D. and N.R.; writing—original draft preparation, C.P., A.R. and G.L.; writing—review and editing, all authors; supervision, F.B., E.B. and G.L.; project administration, G.L.; funding acquisition, F.B., E.B. and G.L. All authors have read and agreed to the published version of the manuscript.

**Funding:** This research was funded by AstraZeneca and INNOV_ECAIR, a unit from the Caen University Hospital Center.

**Institutional Review Board Statement:** The ID-MUT study was approved by the appropriate ethics committee (Avis favorable CLERS ref ID#1595, France). In accordance with the French law n° 2018-493 of 20 June 2018 relating to the protection of personal data, a formally complete declaration file was sent to the National Commission for Data Protection (CNIL; declaration number: 2204611 v 0).

**Informed Consent Statement:** Written informed consent was obtained from the patient(s) to publish this paper.

**Data Availability Statement:** All data are stored at the CHU of Caen and CFB center and can be made available upon request.

**Acknowledgments:** The authors thank all of the onco-pulmonologists and anatomopathologists from Caen University Hospital and the François Baclesse Center that contributed to this study; Sylvie Lecot-Cotigny, Géraldine Lefevre, and Roseline Patey for their technical support in the realization of this study; and Vincent Léon, Clinical Research Associate at Caen University Hospital, for his support in this project.

**Conflicts of Interest:** G.L. received 22 batches of six *EGFR* Idylla^TM tests (Biocartis®) from AstraZeneca. FB received payment for participating in international meetings from Amgen, AstraZeneca, BMS, MSD, and Sanofi. The funders had no role in the design of the study; in the collection, analyses, or interpretation of data; in the writing of the manuscript, or in the decision to publish the results. The other authors declare no conflicts of interest.

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
