# Peer review of "Contribution of the IdyllaTM System to Improving the Therapeutic Care of Patients with NSCLC through Early Screening of EGFR Mutations"

_curroncol, doi:10.3390/curroncol28060376_

Round 1
Reviewer 1 Report
Major comment:
Please explain what causes the delays for Idylla samples for which the time from tumor sampling to genotyping is longer than a couple of days (even more than 15 days for 10% of the samples - Fig 2). If a sample is available it can be directly analysed on the Idylla with a result 2h later.
Page 6 section 3.2: If 25 samples carry an EGFR mutation with 1 sample carrying a double mutation, then the total number of EGFR mutations is 26 and not 27 (line 176). Please check and correct this inconsistency.
Minor comments:
Gene symbols should be in italic.
Revise sentence on page 2 lines 72-76. Line 89, change 'EGFR maturations' to 'EGFR mutations' and '225' to '223'.
Page 3: Lines 120 and 121, change 'variations' to 'variants'. Lines 131 and 133, change 'when found' to 'if present'. Line 135, change 'averaged' to 'ranged'. Line 142, change 'technic varible' to 'used technique'.
Table 1: Stage - numbers do not fit. No % provided.
Page 6: Revise sentence on lines 182-185. Change 'shavings' to 'scraping'. Line 97, EGFR genotyping results were known. Indicate by which method.
I find it confusing to use 2 different time indications (cd and wd). Choose one. Subsequently, remove Fig. 2B or 2C.
Line 206, change 'times periods' to 'time periods'. Lines 223 and 233, change 'for deliver' to 'to deliver'. Line 226, ... of genotyping 'results' ...
Line 183, change 'sand' to 'and'. Line 184, remove 'as a reminder'. Line 309, change 'trues positives' to 'true positieves'.
Author Response
Major comment:
We would like to thank the reviewer #1 for his/her interests and valuable comments on the manuscript.
Please explain what causes the delays for Idylla samples for which the time from tumor sampling to genotyping is longer than a couple of days (even more than 15 days for 10% of the samples - Fig 2). If a sample is available it can be directly analysed on the Idylla with a result 2h later.
We think there may have been some confusion here. The delay between EGFR genotyping prescription (i.e. when histology confirmation of adenocarcinoma is established) is 1.9 ± 1.8 calendar days (or 1.5 ± 1.3 working days) so, no more longer than a couple of days. It seems to us that this reviewer is talking about the global delay from the tumor sampling (biospy) to the rendering of the EGFR genotyping result (9.4 ± 5.4 calendar days or 6.7 ± 3.9 working days). But as the reviewer knows perfectly well, it takes a certain time to allow the fixation of the tumor sample, the production of the slides stained with hematoxylline, eosin and saffron, and then the immunohistochemistry allowing to make the diagnosis of adenocarcinoma, this average delay is 7.4 ± 5.1 calendar days (or 5.3 ± 3.6 working days).
On the other hand, it is true that the prescription of certain analyses by Idylla was delayed due to a lack of training of the prescribers, the time it takes for this habit to be well taken by then.
Page 6 section 3.2: If 25 samples carry an EGFR mutation with 1 sample carrying a double mutation, then the total number of EGFR mutations is 26 and not 27 (line 176). Please check and correct this inconsistency.
We thank the reviewer for pointing this out. The correction has been made.
Minor comments:
Gene symbols should be in italic. Sorry for this oversight, the corrections have been made, the genes are now indicated in italic as the convention requires.
Revise sentence on page 2 lines 72-76. Line 89, change 'EGFR maturations' to 'EGFR mutations' and '225' to '223'. We thank the reviewer for pointing this out. This sentence has been revised.
Page 3: Lines 120 and 121, change 'variations' to 'variants'. Lines 131 and 133, change 'when found' to 'if present'. Line 135, change 'averaged' to 'ranged'. Line 142, change 'technic varible' to 'used technique'. We thank the reviewer for pointing this out. The corrections have been made
Table 1: Stage - numbers do not fit. No % provided. We thank the reviewer for pointing this out. The corrections have been made
Page 6: Revise sentence on lines 182-185. Change 'shavings' to 'scraping'. Change done. Line 97, EGFR genotyping results were known. Indicate by which method. The sentence was modified as following : « For 23/158 patients, EGFR genotyping results were known (10/23 EGFR genotyping results were based only on the IdyllaTM method, 13/23 EGFR genotyping results were known from both NGS and IdyllaTM methods). »
I find it confusing to use 2 different time indications (cd and wd). Choose one. Subsequently, remove Fig. 2B or 2C. We wished to express the delays in working days and calendar days so that this article is readable from the point of view of practitioners (working days) and patients (calendar days) respectively. We believe that this also contributes to the originality of our study. Thus, with all due respect to the reviewer1, we do not wish to apply this recommendation. Nevertheless, we will explain this choice in the manuscript with the sentence « Figure 2B,C illustrate the availability of EGFR genotyping since the tumor sampling in calendar days (Figure 2B) and working days (Figure 2C) to be readable from the point of view of patients (calendar days) and practitioners (working days) respectively » specify in the results section.
Line 206, change 'times periods' to 'time periods'. Lines 223 and 233, change 'for deliver' to 'to deliver'. Line 226, ... of genotyping 'results' ... The corrections have been made
Line 183, change 'sand' to 'and'. Line 184, remove 'as a reminder'. Line 309, change 'trues positives' to 'true positieves'. The corrections have been made
Reviewer 2 Report
Suggestions for small changes in the text:
line 49-50: driver molecular abnormalities: MET exon 14 skipping should be included.
line 63: stage IB to IIIA resected EGFR mutation-positive NSCLC, remove words: EGFR mutation-positive
line 69: state the initiation, should be start?
line 75: insertions in exon 20.
line 89: number of patient is 223, not 225.
line 208: replace EGFR genotyping prescription by EGFR genotyping order/request.
lines 359-360; missing word: two weeks.
Materials and methods:
1. please explain the routinely testing by NGS in terms of frequency of testing. This has an enormous impact on the outcome of the study. The difference between one NGS run a week or two runs a week has a dramatic effect on the timeframe of the genotyping.
2. Since 23% of the samples are cytology specimens, please describe in what way the processing of cytological samples differ from normal FFPE samples.
3. Idylla procedure: As the authors describe in methods, they did not work according the recommendations from Biocartis, regarding the amount and type of input material tested. For instance, because there was a risk of missing out on a EGFR mutation with low allelic frequency, 6-12 times more input material was used. Please discuss the risks of using too much input material.
4. What is the minimal tumor content of the samples, used in this study? The authors should mention in the text who estimated the slides for tumor content. Was macrodissection performed to increase the tumor cell content?
5. lines 138-139: The definitions of specificity and sensitivity are switched.
6. What was the procedure of cutting of the slides for NGS and Idylla? Was it a split sample approach or have the slides for one test all been cut before the slides needed for the other test? (perhaps the order of the cutting: Idylla-NGS-Idylla-NGS etc?)
Results
1. Patients were excluded, based on tumor content. line 158, 159: what is sufficient tumor cells, what is used as lower limit?
2. in 3.2 the seven missed-mutations EGFR are discussed. One of them was missed probably because following the shavings of the FFPE-block for Idylla to analyse, only 5% of the tumor cells remained in the samples. What was the VAF of the mutation of this sample, as detected by NGS? This is more exact than the estimated tumor cell percentage.
3. Fig 2A: Because one would expect a big difference in TAT between an EGFR genotyping request one day before starting the NGS procedure and one day after starting the NGS procedure, the given SD in fig 2A is rather small (3,5 cd (2,0 working days). Could you explain this?
4. Table 2: Title contains two spelling mistakes (NSCLC and EGFR).
5. Table 2: Patient 193 shows a complex EGFR mutation p.(Leu858_Lys860delinsArgAlaIle) that is not represented in Table S1 which contains the mutations detected with the Idylla EGFR mutation cartridge. Still, EGFR-TKI initiation was started based on the Idylla results! The authors should mention that this mutation was in fact detected, although not on the Idylla list.
Discussion
In the materials and methods ( l. 126) and the discussion (l. 289-291), off-label use of the EGFR Idylla test is discussed as a way to get passed the limitation of an EGFR mutation with low allelic frequency. The authors should mention that off-label use of CE-IVD methods should be thoroughly validated before used in routine testing.
Author Response
We would like to thank the reviewer #2 for his/her interests and valuable comments on the manuscript.
Suggestions for small changes in the text:
line 49-50: driver molecular abnormalities: MET exon 14 skipping should be included. We thank the reviewer2 for this relevant remark, the addition was made in the text.
line 63: stage IB to IIIA resected EGFR mutation-positive NSCLC, remove words: EGFR mutation-positive the correction is made
line 69: state the initiation, should be start? We rephrased this confused sentence by « …when these results are necessary for the initiation of treatment of patients”
line 75: insertions in exon 20. the correction is made
line 89: number of patient is 223, not 225. We thank the reviewer for pointing this out. This sentence has been revised.
line 208: replace EGFR genotyping prescription by EGFR genotyping order/request. The corrections have been made
lines 359-360; missing word: two weeks. the correction is made
Materials and methods:
- please explain the routinely testing by NGS in terms of frequency of testing. This has an enormous impact on the outcome of the study. The difference between one NGS run a week or two runs a week has a dramatic effect on the timeframe of the genotyping. The reviewer is right, we have now specified that the NGS run for the CLV3 panel takes place once a week as following: “EGFR mutation testing by NGS method was centralized and carried out once a week in the Department of Genetics from the CHU de Caen”
- Since 23% of the samples are cytology specimens, please describe in what way the processing of cytological samples differ from normal FFPE samples. Thanks to the remark of the reviewers2&3, we now explained how are realized the paraffin-embedded specimens: “All tumor samples were fixed in formalin for 6 to 48 hours: i) “biopsy” type samples (including bone samples) were addressed to the pathology department from the Caen University Hospital or from the CFB in formalin buffered at 4 %, ii) the “fine needle aspiration” type samples were addressed to the pathology departments in a fixing liquid (CytoRich™, BD, France), the cells were pelleted by centrifugation then incubated in 4% buffered formalin. Following the fixation step, the bone samples undergo an additional decalcification step by being incubated in a buffered EDTA solution (Osteosoft™, Merck, Germany). After fixation (and decalcification for bone samples), the tumor samples were embedded with paraffin.”
- Idylla procedure: As the authors describe in methods, they did not work according the recommendations from Biocartis, regarding the amount and type of input material tested. For instance, because there was a risk of missing out on a EGFR mutation with low allelic frequency, 6-12 times more input material was used. Please discuss the risks of using too much input material. To our best knowledge, there was no risk of saturating the EGFR cartridge with this amount (3 sections of 20 µm) of material (no limit was indicated in the technical procedure for these cartridges). Our experience shows us that with this way of proceeding, we only exceptionally have defective EGFR-cartridges. We specify it now in the discussion section as follows: “This did not put the performance of the tests since no analysis failure due to a saturation of a cartridge was reported. As we have an excellent agreement (96.4%) of results between the methods by IdyllaTM system and NGS, we concluded that our procedure for EGFR mutation testing by the IdyllaTM system allowed not to miss a mutation due to lack of sensitivity for the IdyllaTM method.”
- What is the minimal tumor content of the samples, used in this study? The authors should mention in the text who estimated the slides for tumor content. Was macrodissection performed to increase the tumor cell content? We now specify that: "the tumor sample had to contain at least 10% of tumor cells, a macrodissection was carried out to enrich the sample with tumor cells when necessary”
- lines 138-139: The definitions of specificity and sensitivity are switched. We thank the reviewer for pointing this out. The correction is made.
- What was the procedure of cutting of the slides for NGS and Idylla? Was it a split sample approach or have the slides for one test all been cut before the slides needed for the other test? (perhaps the order of the cutting: Idylla-NGS-Idylla-NGS etc?) We always perform the sections of the paraffin-embedded specimens for NGS assay before the sections for Idylla method. It is now explained in the Materials and Methos section as following: “Sections paraffin-embedded specimens for EGFR mutation testing by NGS method were performed before those for EGFR mutation testing by the IdyllaTM system. At the end of the two series of sections, a slide for HES (hematoxyllin, eosin, saffron) staining and morpho-logical verification of the residual tumor material was systematically carried out.”
Results
- Patients were excluded, based on tumor content. line 158, 159: what is sufficient tumor cells, what is used as lower limit? The lower limit was that the tumor sample must contain at least 10% tumor cells as now specified in the Materials and Methods section.
- in 3.2 the seven missed-mutations EGFR are discussed. One of them was missed probably because following the shavings of the FFPE-block for Idylla to analyse, only 5% of the tumor cells remained in the samples. What was the VAF of the mutation of this sample, as detected by NGS? This is more exact than the estimated tumor cell percentage. For this patient, the NGS analysis shows a deletion in exon 19 of the EGFR gene with an allelic frequency of 2.80%, this precision was provided in the text as following " However, one of them was missed probably because following the scraping of the FFPE block for IdyllaTM to analyze, only 5% of the tumor cells remained in the sample (the sensitivity threshold of the IdyllaTM technique is 10% of tumor cells) and the deletion of the exon 19 of the EGFR reported by NGS was with a low allelic frequency of 2.80%."
- Fig 2A: Because one would expect a big difference in TAT between an EGFR genotyping request one day before starting the NGS procedure and one day after starting the NGS procedure, the given SD in fig 2A is rather small (3,5 cd (2,0 working days). Could you explain this? We were also surprised of this low standard deviation, we checked the calculations to be sure there was no error. We believe that these small deviations are possibly explained by the fact that the EGFR genotyping request received one day before starting the NGS procedure will not be treated in this run, but in the next run, since DNA extraction must take place, so this request will be processed within the same timeframe as the EGFR genotyping request received one day after starting the NGS procedure.
- Table 2: Title contains two spelling mistakes (NSCLC and EGFR). The corrections have been made
- Table 2: Patient 193 shows a complex EGFR mutation p.(Leu858_Lys860delinsArgAlaIle) that is not represented in Table S1 which contains the mutations detected with the Idylla EGFR mutation cartridge. Still, EGFR-TKI initiation was started based on the Idylla results! The authors should mention that this mutation was in fact detected, although not on the Idylla list. We thank the reviewer for pointing this out. The correction is made.
Discussion
In the materials and methods ( l. 126) and the discussion (l. 289-291), off-label use of the EGFR Idylla test is discussed as a way to get passed the limitation of an EGFR mutation with low allelic frequency. The authors should mention that off-label use of CE-IVD methods should be thoroughly validated before used in routine testing. We thank the reviewer for pointing this out. We now mention that “The off-label use of CE-IVD methods was thoroughly validated before used in routine testing during a retrospective study not reported here.”
Reviewer 3 Report
The article by Petiteau and al reports the impact of adding a rapid EGFR testing to the current practice of lung carcinoma pathological diagnosis. The study was based on a large series. It demonstrated the benefit of a rapid assessment of EGFR status for initiating an adapted treatment in routine practice. Such benefit had not yet been reported and is an important fact that must be published.
I have only some remarks.
1- Page 3 lines 97-98. Is this sentence relevant? If yes: how many cases were excluded like this? Did really any patient under 18 suffer from lung carcinoma during this study? What were the criteria for a poorly fixed sample?
2- Page 3 line 108. Practical data are lacking for assessing the turnaround time of the genotyping in the authors’ routine practice. The study is from 2 medical centers that are located in the same town. Are NGS tests performed in both centers, or only in one center? How long does it take for samples to be delivered from pathology departments to genetic department(s)? Are departments of pathology and genetics combined in a single department? Are DNA extractions and NGS tests routinely performed each day or only some days in the week?
3- Page 3 paragraph 2.3. Are IdyllaTM systems located in the authors’ departments of pathology?
4- Page 4 table 1. The carcinoid tumor is usually not considered a non-small cell lung carcinoma. This case could be deleted, as had been the case with “malpighia metaplasia” (page 5 line 159).
5- Page 6 paragraph 3.2. The authors compared EGFR genotyping by IdyllaTM and NGS methods. They present the comparison as if any difference was an Idylla false result. This comparison must be objective and reported more fairly since NGS methods may also fail. If both medical centers have a department of molecular genetics, discordant cases could easily be assessed by a third analysis.
6- Page 6 line 185. Had the bone metastasis sample been decalcified? Please specify.
7- Page 6 paragraph 3.4. The turnaround times of the 2 centers are not reported separately. Were they compared?
Author Response
The article by Petiteau and al reports the impact of adding a rapid EGFR testing to the current practice of lung carcinoma pathological diagnosis. The study was based on a large series. It demonstrated the benefit of a rapid assessment of EGFR status for initiating an adapted treatment in routine practice. Such benefit had not yet been reported and is an important fact that must be published.
We would like to thank the reviewer #3 for his/her interests and valuable comments on the manuscript.
I have only some remarks.
1- Page 3 lines 97-98. Is this sentence relevant? If yes: how many cases were excluded like this? Did really any patient under 18 suffer from lung carcinoma during this study? What were the criteria for a poorly fixed sample? The reviewer3 is right, the sentence was discarded, since no patient was excluded on the base of these criteria.
2- Page 3 line 108. Practical data are lacking for assessing the turnaround time of the genotyping in the authors’ routine practice. The study is from 2 medical centers that are located in the same town. Are NGS tests performed in both centers, or only in one center?
We thank the reviewer for pointing this out. We answered these questions by specifying in the text of our manuscript (please see the Materials and Methods section):
“Sections paraffin-embedded specimens for EGFR mutation testing by NGS method were performed before those for EGFR mutation testing by the IdyllaTM system. At the end of the two series of sections, a slide for HES (hematoxyllin, eosin, saffron) staining and morpho-logical verification of the residual tumor material was systematically carried out.”
(…)
“EGFR mutation testing by NGS method was centralized and carried out once a week in the Department of Genetics from the CHU de Caen. Tumor genomic DNA was extracted from three sections of 10 µm of paraffin-embedded specimens using an RSC FFPE Plus DNA Kit (Promega™) on a Maxwell RSC 48 auto-mated system (Promega™) according to the manufacturer’s recommendation twice a week.”
(…)
“EGFR mutation testing by the IdyllaTM system was centralized and carried out every working day in the Department of Pathology from the CHU de Caen”
How long does it take for samples to be delivered from pathology departments to genetic department(s)? Samples are delivered the day of the request. Are departments of pathology and genetics combined in a single department? No, they are two different departments but coordinated because both members of a federative structure Federative structure of cyto-MOlecular oncogenetics of the CHU of Caen (SF-MOCAE)
3- Page 3 paragraph 2.3. Are IdyllaTM systems located in the authors’ departments of pathology? Yes, the materials and methods section now mentions that “EGFR mutation testing by the IdyllaTM system was centralized and carried out every working day in the Department of Pathology from the CHU de Caen”
4- Page 4 table 1. The carcinoid tumor is usually not considered a non-small cell lung carcinoma. This case could be deleted, as had been the case with “malpighia metaplasia” (page 5 line 159). We agree with the reviewer3 but for these patients with particular clinical data (smoker ..), the oncologists requested a search for mutation of EGFR which was treated the request of patients with NSCLC and adenocarcinoma. Thus, with all the respect we have for reviewer3, we do not prefer to take his/her recommendation into account and keep these patients in the ID-MUT cohort. We hope reviewer3 will understand our point of view.
5- Page 6 paragraph 3.2. The authors compared EGFR genotyping by IdyllaTM and NGS methods. They present the comparison as if any difference was an Idylla false result. This comparison must be objective and reported more fairly since NGS methods may also fail. If both medical centers have a department of molecular genetics, discordant cases could easily be assessed by a third analysis. The reviewer3 is right, but it turns out that concerning the samples of this study, there was no false positive in the NGS analysis. In the future, if this were to be the case, we would verify such discordant by Sanger sequencing, which has the same sensitivity (10% tumor cells as the Idylla method) or droplet digital PCR.
6- Page 6 line 185. Had the bone metastasis sample been decalcified? Please specify. Thanks to the remark of the reviewers2&3, we now explained how are realized the paraffin-embedded specimens: “All tumor samples were fixed in formalin for 6 to 48 hours: i) “biopsy” type samples (including bone samples) were addressed to the pathology department from the Caen University Hospital or from the CFB in formalin buffered at 4 %, ii) the “fine needle aspiration” type samples were addressed to the pathology departments in a fixing liquid (CytoRich™, BD, France), the cells were pelleted by centrifugation then incubated in 4% buffered formalin. Following the fixation step, the bone samples undergo an additional decalcification step by being incubated in a buffered EDTA solution (Osteosoft™, Merck, Germany). After fixation (and decalcification for bone samples), the tumor samples were embedded with paraffin.”
7- Page 6 paragraph 3.4. The turnaround times of the 2 centers are not reported separately. Were they compared? Yes, the turnaround times according to the two centers were tested, there was no significant difference between them. We now indicate this notion in the text, at the end of the presentation of these turnaround times: « All these TAT (timeframe between (1) tumor sampling and EGFR genotyping prescriptionrequest, (2) EGFR genotyping prescription request and result, (3) tumoral sampling and EGFR geno-typing result, (4) results from both techniques, and (5) tumoral sampling and initiation of treatment) were comparable between the CHU and the CFB (p>0.05). »